# Quantum Dot Imaging Agents: Haematopoietic Cell Interactions and Biocompatibility

**DOI:** 10.3390/cells13040354

**Published:** 2024-02-18

**Authors:** Leigh Naylor-Adamson, Thomas W. Price, Zoe Booth, Graeme J. Stasiuk, Simon D. J. Calaminus

**Affiliations:** 1Centre for Biomedicine, Hull York Medical School, University of Hull, Hull HU6 7RX, UK; 2Department of Imaging Chemistry and Biology, School of Biomedical Engineering and Imaging Sciences, King’s College London, London SE1 7EH, UK

**Keywords:** quantum dots, cardiovascular diseases, imaging

## Abstract

Quantum dots (QDs) are semi-conducting nanoparticles that have been developed for a range of biological and non-biological functions. They can be tuned to multiple different emission wavelengths and can have significant benefits over other fluorescent systems. Many studies have utilised QDs with a cadmium-based core; however, these QDs have since been shown to have poor biological compatibility. Therefore, other QDs, such as indium phosphide QDs, have been developed. These QDs retain excellent fluorescent intensity and tunability but are thought to have elevated biological compatibility. Herein we discuss the applicability of a range of QDs to the cardiovascular system. Key disease states such as myocardial infarction and stroke are associated with cardiovascular disease (CVD), and there is an opportunity to improve clinical imaging to aide clinical outcomes for these disease states. QDs offer potential clinical benefits given their ability to perform multiple functions, such as carry an imaging agent, a therapy, and a targeting motif. Two key cell types associated with CVD are platelets and immune cells. Both cell types play key roles in establishing an inflammatory environment within CVD, and as such aid the formation of pathological thrombi. However, it is unclear at present how and with which cell types QDs interact, and if they potentially drive unwanted changes or activation of these cell types. Therefore, although QDs show great promise for boosting imaging capability, further work needs to be completed to fully understand their biological compatibility.

## 1. Introduction

Since their first description in 1981 [1], quantum dots (QDs) have been developed for a range of different applications, from LED screens and energy conservation to biological applications, with the first known use in biological imaging dating back to 1998 [2]. QDs are inorganic semiconducting nanocrystals. They are a form of nanoparticle but differ from other nanoparticles such as liposomes or dendrimers due to their tunable emission spectra (Figure 1).

The optical properties can be changed by size (QDs are usually between 1 and 10 nm), offering benefits over traditional fluorescent dyes. The smaller the size, the higher the band energy gap, which will serve to shorten the emission wavelength due to the higher energy of the emission photon. The larger the size, the lower the band gap and a decrease in emission photon energy is observed resulting in an increased wavelength. This allows the QDs to emit from the ultraviolet (UV) to the infrared (IR) wavelength. A reduction in optical overlap and therefore better detection sensitivity is also achieved by the large Stokes shift of the QDs. Other benefits of QDs include: the ability to form multiplexes due to a high surface area to volume ratio, which would allow for multimodal probe attachment; photostability equating to no loss of signal for large time periods; the possibility of multiple fluorescent colours to be simultaneously excited; a high signal; and a high brightness, which leads to improved sensitivity due to their large range of excitation [4,5,6,7].

The most common type of QD studied have a cadmium-based core, for example cadmium selenide (CdSe), or cadmium telluride (CdTe). Cadmium QDs have good optical properties and typically easier synthesis, and as such there is great interest for their use [3]. However, critical to the use of QDs within a biological system is the need for the QD to be biologically compatible. This is a significant issue with cadmium based QDs, partly due to the release of cadmium ions (Cd^2+^) and the generation of reactive oxygen species (ROS) [8,9,10,11,12]. Therefore, different types of QDs have been developed with significant interest in indium based QDs (indium-phosphide (InP)) [13]. In addition to using a different metal core, the QD core can be coated with a semiconducting shell, frequently zinc sulphide (ZnS), which improves stability. From here, the QD can be capped with polymer layers or biocompatible materials, which enhances the QD bioactivity as well as solubility in water [14]. If coating with a polymer, the use of -COOH and -NH_2_ as surface groups allows for the QD to be covalently coupled to other molecules [15]. Furthermore, the functional groups in the QD shell can be used to covalently attach specific ligands. This can also be achieved through bioconjugation between streptavidin–biotin. These ligands can utilise the QD for specific purposes to provide applications in targeting, diagnostics, and therapeutics [3].

Biological imaging often relies on fluorescence or chemiluminescence; however, visible light has poor transmission through tissue due to high light scattering and tissue absorption, leading to limitations in animal models [16]. Here, the QD’s emission spectrum is beneficial to allow for deep tissue imaging [17] as the lower light scattering and absorption allows for a high resolution. Indium arsenide (InAs) QDs emit in the short-wavelength IR (700–1200 nm) and are particularly useful due to high sensitivity and deep signal penetration in vivo. They have been used to image blood flow in the brain vasculature in mice with glioblastoma tumours. Three days before imaging, the tumour was labelled by QDs which were known to accumulate in abnormal blood vessels. Before imaging, a QD composite particle was injected into the mice allowing for high-speed, real-time imaging and thus the visualisation of the vascular abnormalities of the tumour. This process also permitted the flow of the vasculature in the healthy tissue versus the tumour margin, the data from which, due to high resolution, could be used to generate flow maps as a quantitative measurement of large areas of mapped brain sections. This technique is proposed to be useful in the future for a multitude of brain pathologies, including stroke. These QDs were found after 6 days to induce no hepatotoxicity or nephrotoxicity, and full blood counts suggest no changes in leukocyte, red blood cell (RBC), platelet, or haemoglobin levels [18]. Deeper tissue penetration can also be achieved by utilising X-rays to excite near-infrared (NIR) CdTe QDs to image the liver, kidneys, and brain of mice [19].

However, while still providing good temporal and spatial resolution, QDs can have issues with stability in an aqueous solution and can exhibit low brightness. Subsequently, QD probes have been developed to overcome these problems. This has been seen with lead sulphide (PbS) QDs, synthesised with a cadmium sulphide (CdS) shell. The use of the CdS shell acts as an amphiphilic polymer coating, allowing for aqueous stability, as well as protecting the PbS core from oxidation, and therefore maintaining a bright fluorescence. The QDs were coated in Polyethylene Glycol (PEG) and allowed the imaging of mouse blood flow at the high rate of 60 fps, as well as imaging of mouse tumour vasculature. Although the QDs use Pb, a known toxic element, the QDs were found to be effectively cleared from the body and organs, with only 0.7% remaining after 28 days, with mice showing no weight loss or death, and therefore no obvious toxicity [20].

In addition to good tissue penetration, the QD size can be changed alongside the core materials to create a range of emission spectra from UV emitting graphene QDs [21], near UV emitting zinc selenide (ZnSe) QDs fabricated with Europium [22], visible with CdTe QDs with ZnSe shells [23], and finally to near IR, as seen with PbS/CdS QDs [24]. Additionally, QDs can be used for multimodal imaging through the attachment of contrast agents for magnetic resonance imaging (MRI), radioisotopes for either positron emission tomography (PET), or single photon emission computed tomography (SPECT) [25,26,27]. Furthermore, there is the potential for QDs to be utilised as drug delivery systems. However, the biocompatibility of QDs in the cardiovascular and haemopoietic systems is a matter of concern, especially in regard to the unwanted interactions with specific haematopoeitic cell types.

## 2. Cardiovascular System

Biological imaging is key to aiding the clinical understanding of the cardiovascular system. An effective cardiovascular system is critical to a healthy body, as the effective movement of blood delivers substances such as glucose and oxygen to tissues, but also removes metabolites ready for excretion. However, the establishment of cardiovascular disease (CVD), a term that covers diseases such as coronary artery disease, acute coronary syndrome, atherosclerosis, and ischemic heart disease, causes significant change to the cardiovascular system. These changes can ultimately lead to significant rates of morbidity and mortality due to events such as myocardial infarction and stroke [28]. Such events require effective and timely treatment, and imaging is a widely used tool to aid the medical treatment of these conditions. Currently, imaging of the heart is conducted through tomography, echocardiogram, and MRI [29]. These methods, however, all leave room for improvement in terms of spatial resolution, as this would improve insights into the local environment such as local myocardial movements. More recently, optical coherence tomography has been found to be a potential tool for high-resolution imaging of the cardiovascular system, but current contrast agents prevent the full potential of this imaging technique [30]. Advancement of vascular imaging could therefore aid in diagnosis and therapeutic options.

One key disease process underpinning CVD is atherosclerosis. This disease is caused by the formation of fibrofatty lesions on artery walls that can cause the narrowing of blood vessels, activation of inflammatory pathways, and results in the formation of a plaque. These plaques can become unstable and rupture, leading to acute events such as myocardial infarction or stroke [31]. Atherosclerosis has been targeted using QDs by utilising different methodologies (Figure 2).

Selenium, which is a possible nanomaterial for QDs, has been reported to help prevent atherosclerosis [32]. Due to the narrow safe range of concentration of selenium, selenium QDs (SeQDs) were developed as an alternative, potentially safer formulation. These have been shown to reduce endothelium dysfunction in rats fed a high fat diet, particularly in comparison to selenium supplemented in the diet, as well as reduced restenosis. The SeQDs were found to inactivate Na^+^/H^+^ exchanger 1, preventing the growths of plaques in ApoE^−/−^ mice [32].

PbS QDs functionalised with a PEG ligand with an emission of 1600 nm have been designed to aid in intracoronary optical coherence tomography, with the benefit of IR contrast which can be stimulated under the same laser excitation. These QDs worked well in colloidal suspension simulating an artery, as well as in tissue. Ex vivo experiments in rabbit aorta were able to provide proof of concept, with simultaneous IR and optical coherence tomography providing accurate artery morphology. There is the potential for these QDs to accumulate at sites of damaged endothelium and atherosclerosis to provide a future diagnostic tool [30].

Alternatively, QDs have been used to therapeutically target atherosclerotic plaques. Simian Virus 40 (SV40) virus-like particles have been utilised due to their potential for the addition of multivalent structures. To the SV40 virus, a peptide was added targeting macrophages, chosen due to their association with inflammation and plaque formation, as well as Hirulog peptide, a thrombin peptide inhibitor, to form a pentamer. This pentamer was then assembled with CdSe/ZnS QDs emitting at 800 nm, so that twelve pentamers encapsulated one QD. These QDs were then injected into ApoE^−/−^ mice. The QDs were able to successfully image plaques within the mice, as well as provide selective anticoagulant activity due to the Hirulog peptide, which was especially seen in the aortas; however, changes to the size of plaques were not determined [33].

However, a significant issue to using QDs as imaging agents in vivo is to understand the cellular toxicity that they can cause to the haematopoietic system. QDs have been studied in terms of embryonic development in which embryonic zebrafish were exposed to amino or carboxyl modified CdSe/ZnS QDs. Though the QDs affected survival and hatching rate, interestingly after hatching, the QDs were found to attach to the heart. These QDs were observed to cause cardiac dysfunction and pericardial oedema, with upregulation of cardiac development-related gene transcription, with the carboxyl-QD showing higher levels of toxicity [34]. An invertebrate model, Bombyx mori, has further shown that CdTe QDs affect haematopoiesis, with damage to the haematopoietic organs and haematocytes. This was thought to be down to the large generation of ROS upon exposure to the QDs, as well as another unknown mechanism [11]. This highlights the importance of investigating the potentially lethal effects of QDs in biological systems for the safety of future applications.

Overall, QDs seem to hold much promise in the imaging of CVD, from cadmium QDs to PEGylated PbS QDs, allowing for greater imaging capabilities of arteries to aid in diagnostics in atherosclerosis [30]. Great potential is also seen for QDs in CVD therapeutics, with the delivery of selenium in SeQDs preventing plaque growth [32].

## 3. Hematopoietic System

Although there is a potential promise in the use of QDs for CVD, especially atherosclerosis, it must be noted within the blood there are a range of different haematopoietic cells in terms of number, size, lifespan, and function. Although the ability of these cells to interact with QDs will be cell-type specific, all blood cells could be affected by interaction with QDs. For example, QD interactions with atherosclerotic lesions have been studied in ApoE mice with differently functionalised QDs. Carboxyl-CdSe/ZnS QDs were found to interact with plaques to a quicker and stronger extent than PEG- or amine/PEG QDs, particularly downstream of the lesion, through interactions with platelets and leukocytes. The carboxyl-QDs were also associated with a greater binding to inflammatory blood cells. The interactions with the QDs and the lesion were thought to occur through fibrinogen, and preincubation of PEG-QDs with fibrinogen resulted in the stronger binding of these treated QDs with the downstream region of the lesion. The results from this study serve to highlight the multiple interactions the QDs can have with blood cells as well as coagulation factors and how this can be altered by the many makeups of QDs [35]. In addition, it is not just the cell characteristics, but also the size, charge, and ability to interact with plasma proteins of the QD itself which can affect the QDs biological compatibility. Therefore, there are a range of different variables which can then affect how the QD may react and the types of potential side effects that may occur within the cardiovascular system.

### 3.1. Direct Platelet—QD Interactions

Platelets are small, anucleated cells found only in mammals which form haemopoietic plugs upon blood vessel injury. They live for 7–14 days, and a normal platelet count is between 150 and 400 × 10^9^/L [36]. In addition to their role in preventing bleeding, platelets are known to help co-ordinate the immune system during inflammation [37,38,39]. As they have both FcγRIIa, and damage-associated molecular pattern (DAMP) receptors, they can engulf foreign particles such as bacteria and viruses [40], and thus have the potential to engulf particles such as QDs.

CdTe QDs have been observed to activate platelets. It was found that smaller 2.6 nm QDs were able to cause platelet aggregation at a lower concentration than the larger 4.8 nm QDs (100 nM and 1000 nM respectively), which was thought to be due to a greater surface-to-volume ratio, allowing for more interactions with platelets when the QDs are smaller in size. The study proposed that QDs interact with the integrin α_IIb_β_3_, resulting in platelet activation. An increase in P-selectin, a marker of granule release and MMP-2, a stimulator of aggregation, was also seen in these samples [41].

Studies into the effects of InP/ZnS QDs on platelets have also been conducted. These QDs, were investigated as they have been shown both in vitro and in vivo to be more biologically compatible than Cd QDs [42,43,44]. The QDs were coated with either thioglycolic acid (TGA), lipoic acid (LA), glutathione (GSH), or penicillamine (Pen) in order to be transferred to the aqueous phase from the organic phase. Platelets interacted with the QDs at a low concentration, but there was little effect on function observed. However, increases in QD concentration resulted in platelet activation and aggregation, although this was weaker with TGA coatings. Platelets spread on fibrinogen after incubation with QDs were seen to have normal adherence whilst surface area was decreased with high concentration of LA and GSH coatings. The study concluded that both the concentration and coating of QDs have differing effects on platelets and this needs to be considered before implementation into a biological system [14].

Highlighted by the above studies, more research should be conducted into the role of QDs in platelet activation and the potential implications it has on these cells, including the effects of different coatings and functionalisation. Owing to the multifaceted role of platelets, changes to their activation and status in the cardiovascular system could induce significant side effects, thus QD biocompatibility needs to be ensured.

### 3.2. QD Effects on the Coagulation System

Alongside the platelet, the coagulation cascade is critical in the prevention of bleeding. This cascade is a series of reactions which allow for the formation of a fibrin meshwork, which is insoluble and therefore acts as a scaffold to bind together the thrombus. Unwanted activation of the coagulation cascade can lead to the formation of pathological thrombi, which can block arteries in situ, or can move through the body leading to blood vessel blockage at a secondary site.

The coagulation cascade has been shown to be affected by a range of different QDs [45,46,47]. Carboxyl functionalised CdSe/ZnS QDs were observed to cause thrombosis in the pulmonary vasculature in a dose-dependent manner. In contrast, this was not seen to the same extent in amine-functionalised QDs. This was potentially due to the charge status of the QDs, with amine-functionalised QDs having a zeta potential of −14.2 mV in comparison to the carboxyl QD zeta potential of −35.2 mV. The thrombosis was mediated via activation of the coagulation cascade, as platelet aggregation was not induced in vitro by the QDs, whilst addition of heparin in vivo prevented thrombosis. This indicated that the coagulation cascade was activated by the negative charge on the QDs, which in turn caused platelet activation through the formation of fibrin and the presence of thrombin [45].

CdTe/ZnS QDs can interact and form conjugates with fibrinogen, plasminogen, and prothrombin at a higher affinity than CdTe QDs. The proteins interacting with the QDs were seen to undergo conformational changes, with fibrinogen and prothrombin increasing in α-helices suggesting refolding of the proteins, whilst plasminogen decreased in random coils and α-helices, suggesting unfolding and exposure of previously folded amino acids. In rats, these QDs increased activated partial thromboplastin time and prothrombin time within the first three days after exposure. After day three, coagulation was seen to recover; however, fibrinolysis was still active, as seen by levels of active coagulation factor X, plasminogen, and tissue plasminogen activator [47].

CdTe QDs also have a size-dependent anticoagulant effect. It was demonstrated that whilst 3.6 nm QDs did not have an effect, 3.2 nm QDs affected the intrinsic coagulation pathway as a whole due to an increased thrombin time and activated partial thromboplastin time, though they did not affect the coagulation factors involved within the pathway. The study theorises that the smaller QDs can bind and cause changes in the coagulation factors, whilst the larger QDs size prevents this [46].

From the above studies, it would seem smaller QDs with a more negative zeta potential cause dysregulation of the coagulation system in a manner which may also be dependent upon QD coating. Due to the potentially harmful effects of a dysregulated coagulation system, much more research needs to be completed to ascertain the full effects of QDs on the various proteins of the coagulation system. This research will need to delve into the effects of QD coating, as well as size and zeta potential to find the most biologically compatible QD for future applications.

### 3.3. Red Blood Cells

Red blood cells (RBCs) are another key cell that transport gases and nutrients throughout the body, therefore disruption of their function can have critical effects on the body. This is in part because they have a life span of approximately 120 days, and as such have a slow turnover [48].

The effect of graphene QDs on RBCs has been explored. Kim et al. investigated the effect of non-functionalised, hydroxylated, and carboxylated graphene QDs on RBC function. Rheological changes and haemolysis were insignificant at low concentrations (250 μg/mL). However, at higher concentrations (750 μg/mL), carboxylated graphene QDs were seen to produce deformation of the RBCs, as well as haemolysis and aggregation [49]. Importantly, RBCs, which are incapable of endocytosis, have also been found to have internalised QDs, as the QDs are able to penetrate the lipid bilayer. This penetration and internalisation have been found to increase flexibility of the bilayer which remains intact. This demonstrates that QDs will not just be present within cells that undergo significant amounts of endocytosis [50].

However, importantly RBCs can be adapted to act as a carrier agent for QDs [51]. Hydrophobic CdSe/ZnS QDs functionalised with octadecylamine have been encapsulated within RBC membranes. These encapsulated QDs were then able to fuse with the membranes of other living cells which would be effectively labelled. RBCs were chosen to be membrane lipid donors due to the ease at which they lyse, they lack other types of membrane which prevent contamination, whilst still being structurally similar to other cell types. It was found that small QDs were able to stably label cell membranes both for imaging and single-nanoparticle tracking [51]. In addition, RBC membranes have been identified as a possible mechanism to aid QD chemotherapy. Black phosphorous QDs, which were functionalised with the chemotherapy agent doxorubicin and the anti-inflammatory kirenol, were encapsulated within RBC membranes. These QDs were able to achieve high drug loading and monodispersion capacity due to the RBC membrane use, as well as inhibiting death resistance in tumour cells and downregulated proinflammatory cytokines. Due to use of RBC membranes, the toxicity of doxorubicin alone was reduced [52].

Due to the vital role RBCs play in the cardiovascular system, it is of concern that high concentrations of QDs can permeate into the cells and cause deformation. Further studies into the effects of the QDs on RBCs are required, particularly to visualise the potential effects on a more physiological model. However, the use of RBC membranes as a carrier for QDs for future applications is of interest.

### 3.4. Leucocytes

White blood cells are a key mechanism to control infection. There are multiple types of white blood cells: macrophages, neutrophils, lymphocytes, and T cells. One of the key aspects of white blood cells, especially macrophages, is their ability to engulf foreign particles, such as bacteria [53]. Therefore, it is likely that QDs injected into a biological system could be engulfed by white blood cells. Therefore, ensuring that the QDs do not affect immune cell function, either in terms of preventing activation or inducing over activation, is critical.

It has been proposed that targeting Monocyte-derived macrophages (MDMs) with CdSe/CdZnS QDs inserted into liposomes could be a potential imaging agent to aid diagnosis. MDMs have been found to be increased in the arterial wall in poor prognosis cases of cardiovascular disease with vascular inflammation. CdSe/CdZnS QDs inserted into liposomes were synthesised and used to target the MDMs through local administration to the site of inflammation in rat models of restenosis. Liposome use enabled monocytes and macrophages to internalise the QD. These QDs had good retention over 24 h and had specific uptake into MDMs. They were also found to remain within the vessel walls, providing future potential to act as an intra-arterial therapeutic delivery system. However, significant cytotoxicity of free liposome and QD-liposomes was found [54].

The coating of the QD results in different immune cell interactions. CdSe/ZnS QDs coated with mercaptopropionic acid (MPA), with an emission at 620 nm, 10 nm size, and a zeta potential of −18.2 mV, were found in higher proportion in monocytic, lymphocytic and granulocytic blood fractions. Neutrophils were able to internalise these QDs, and to destroy the QD potentially through the action of ROS. This altered neutrophil morphology and resulted in cell death. The destruction of the QD and the potential release of the toxic shell was indicated to be the reason for neutrophil cell death. CdSe/CdSeZnS/ZnS-polyT QDs (PEGylated) with an emission of 588 nm, 18 nm size, and a zeta potential of −18.57 mV had less of an effect on neutrophils, with almost half the toxicity (LD_50_ 0.025 mg/mL versus 0.04 mg/mL) [55].

Interactions between the lymphocytes and neutrophilic granulocytes (NG) and CdSe QDs were seen to be dependent on the functionalisation of the QD surface. CdSe/Zn-MPA resulted in most interactions with lymphocytes, whilst CdSe/CdZnS/ZnS coated with Polyvinylpyrrolidone heterobifunctional polymer (PTVP) QDs had interactions impeded, and CdSe/CdZnS/ZnS-PTVP-APS QDs had no interactions at all. Importantly there was a difference in the types of interaction, with NGs phagocytosing the QD whilst lymphocytes fixed the QD to their surface. Lymphocytes were observed to be deformed after fixation to QDs. They also showed levelling of receptor sites and perforations appearing as incubation time increased alongside thinning of the fixation site sublayer. For NGs, membrane thinning was also seen, with cell debris seen at sublayer sites. This highlights the importance of looking into different functionalisation of QDs on different cell types [56].

Importantly, QDs do not just interact, they can also alter immune cell function. QDs have been seen to have proinflammatory immune responses. CdSe/ZnS QDs in rats have been shown to increase levels of innate immune cells and adaptive lymphocytes, as well as increased pulmonary inflammatory chemokines. In contrast, mouse macrophages internalise CdSe/ZnS QDs with an uptake in the region of 90%, with decreased viability, increased tumour necrosis factor α (TNF-α) and Interleukin-6 (IL-6) gene transcription [57,58]. Graphene QDs, which were conjugated to PEG, were demonstrated under irradiation to remove squamous cell carcinoma. However, at the same time, proinflammatory cytokines, such as TNF-α and Interferon-γ (IFN-γ) were increased, as were cytotoxic T cells. However, the study proposed the increase in host proinflammatory immunity could positively target immune cells to tumours as a form of immunotherapy [59].

However, other QDs can cause a reduced inflammatory response. Mice with concanavalin A induced hepatitis, a liver injury which is T cell mediated, were injected with graphene QDs which reduced liver damage by decreased autophagy and apoptosis through the QDs reduction in apoptotic markers and mRNA apoptotic mediators. The livers also showed less infiltration of immune cells; of note were T cells producing IFN-γ. These QDs were also demonstrated in vitro to prevent free radical Nitric Oxide (NO) production from macrophages. Overall, this relieved the immune-mediated hepatitis [60]. Graphene QDs have also been shown to change M1 polarisation of macrophages to M2, and to prevent the TH1/TH17 polarisation in acute and chronic colitis as a potential anti-inflammatory therapy [61]. CdSe/ZnS QDs have also been shown to decrease viability and increased apoptotic events of macrophages, increased ROS production and decreased IL-6 and TNF-α release. In lymphocytes, these QDs have increased TNF-α and IL-6 release, an increased viability and a reduced transformation. In BALB/c mice lymphocytes, while IL-6 and TNF-α release was still impaired, cell viability was also decreased. The QDs suppression of the host immune defense is a potential consideration for future use due to increased sensitivity to disease [62].

It is of note that nanoparticles, including QDs, undergo the formation of a biocorona in a biological system. The biocorona is formed as the biomolecules, such as proteins, lipids, and metabolites interact with and coat the surface of the nanoparticle. This biocorona formation is known to change the nanoparticles physiochemical properties [63,64,65,66]. The exact composition of these biocorona then affects how the QD interacts with cells. The route of internalisation of nanoparticles by macrophages is dependent on the biocorona, with nanoparticles without a biocorona being internalised via endocytosis, whilst nanoparticles with biocorona being internalised via phagocytosis [67]. It has previously been found that the biocorona induced by silver nanoparticles in cholesterol-rich media resulted in an increased internalisation by macrophages, as well as changes to scavenger receptor expression in these cells [68]. Black phosphorous QDs have been found to form a biocorona with up to 75.8% of proteins being immune related, with changes to the QD shape, and a decrease in zeta potential [69]. Studies have been performed to elucidate the roles of nanoparticle surfaces on biocorona formation [65,70,71]; however, more needs to be conducted to fully understand the potential implication of biocorona formation on QDs with immune cells, particularly macrophages.

As changes to leucocytes in terms of the morphology, activation, or dampening of responses can have such a large effect on the biological system as a whole, it is vital that interactions of these cells and their consequences are considered. As studies also show differing views on inflammatory effects on QDs, either pro- or anti-, further research is needed to understand the different responses to QDs of different size, type, and zeta potential.

### 3.5. Bone Marrow Stem Cells

Cells of the bone marrow include mesenchymal stem cells (MSCs) and haematopoietic stem cells (HSCs). MSCs generate cells of the mesenchymal lineage, including adipocytes and muscle cells, whilst HSCs generate cells of the haematopoietic lineage, such as leucocytes, RBCs and platelets [72].

HSCs are known to reside in a specialised environment, termed a niche, which determines their use [73,74]. However, there are gaps in the knowledge pertaining to bone marrow niches. To further understand the bone marrow niche, QDs, either CdSe/ZnS (655 nm), or CdTe/ZnS (800 nm), were injected into mice retro-orbitally to allow circulation in the vasculature and hence visualisation of the bone marrow vasculature. Upon irradiation, QDs were no longer able to visualise the bone marrow vasculature, instead fluorescing in the cavity with days from irradiation showing progressive damage, though blood cells seemed to remain in the vasculature. This leads to the hypothesis that bone marrow is permeable to small particles such as QDs after irradiation. However, this technique was seen to have limits in the QD detection, with depths of over 60 μm in the endosteal surface the vascular connective tissue membrane, seeing only a faint QD signal which could not be analysed [75].

HSC transplantation is a tool which has potential in disease. QDs have been developed to observe the effects of HSC transplantation. CdSe/ZnS QDs were coated with 1,2-distearoyl-sn-glycero-3-phosphoetholamine (DSPE)-PEG 2000 amine and further conjugated with trans-acting activator of transcription (Tat) peptide. Tat peptide is a cell-penetrating peptide used frequently to transport molecules into cells. QDs were incubated with HSCs and confocal microscopy was used to verify uptake of QD into the cells, with saturation found to occur at four hours. HSCs labelled with red QDs were injected into the tail veins of mice, which were killed after 24 h. HSCs were detected in the lungs of these mice, as well as in the liver and spleen. The authors postulated that this could have application in the modulation of HSC homing [76].

Intra-cellular delivery of QDs is more challenging than the targeting to surface proteins, though there are several ways which this can be accomplished, such as lipid-based transduction [77], microinjection [78], and electroporation [79]. One study looking into QD intra-cellular delivery via a peptide-based Qtracker system, in which a peptide is used to deliver the QD to the cytoplasm of cells, assessed both cell type and QD size on QD toxicity in HSCs and MSCs. QDs with wavelengths of 525 and 585 nm were CdSe/ZnS, whilst QDs of wavelength 800 nm were CdTe/ZnS. It was found that the QDs had no toxicity effects on the cells. Both HSCs and MSCs were able to differentiate into multiple lineages as expected with no morphological changes. CD133^+^ and CD34^+^ cells saw a loss of fluorescence in 24 h, potentially due to exocytosis of the QDs. In CD14^+^ cells and MSCs, QDs were more stable over time, with the stability dependent on QD size. However, in animal models and co-cultures, QDs were able to leave the labelled cells to enter other cells. The study concluded that whilst QDs could efficiently label the HSCs and MSCs, the lack of stability in smaller-sized QDs coupled with the leakage of all QDs to other cells are important considerations for future work [80].

Though the use of QDs could help further understand HSCs in their niche as well as in transplantation, more research is needed to develop these probes. Whilst it is promising that certain QDs have shown no toxicity, changes to morphology, or changes to differentiation in HSCs, future work should concentrate on assessing QD variants on toxicity, as well as increasing stability and reducing leakage to other cells.

### 3.6. QD Imaging of Haematopoietic Cells

Previously haemopoietic cells have been both internally and externally labelled for imaging. External marking can be achieved by targeting the QD to surface markers on the cell via antibodies or streptavidin-biotin system. This has been achieved using the Epidermal growth factor receptor (EGFR) in HeLa cells, neuron a -amino-3-hydroxy-5-methyl-4-isoxazolepropionate (AMPA) receptors [81,82,83], CD45, CD5, and CD19, to overall allow imaging via fluorescent microscopy and flow cytometry [84]. Internal markers can be used for internal cell imaging, with more recent work furthering the ability for QDs to be utilised in imaging live cells. The pH ranges in cells can also be imaged in real time by QDs such as CdSe/ZnS QDs functionalized with D-penicillamine and histidine for use in fluorescent lifetime imaging microscopy. These QDs showed a change in fluorescent intensity and lifetime depending on the pH of the cell environment, decreasing in more acidic conditions [81].

Due to the use of QDs to label cells and the QDs ability to emit at different wavelengths, Jayagopal et al. used cadmium QDs to differentiate different cell types to understand the composition of atherosclerotic plaques. The QDs with emissions at 585 and 655 nm were functionalised with biotinylated maurocalcine to penetrate monocytes and T cells for efficient loading with no identified effect on viability or function. The QDs also showed a high signal-to-noise ratio with signal lasting for a month before fading. Localisation of these labelled cells were observed in ApoE^−/−^ mice within 2 days, with different localisation and accumulation patterns dependent on the age of the mouse [85].

Cell membranes can be labelled with QDs, with the capability for membrane potential to be imaged. This has been completed by CdSe/ZnS QDs capped with glutathione which, at neutral pH can label the membrane and in combination with dipicrylamide, in a process similar to fluorescence resonance energy transfer (FRET), can visualise membrane potential by fluorescence [86]. Membrane diffusion and localisation of proteins can also be imaged, as seen in one study of a dopamine transporter using CdSe/ZnS QDs conjugated to streptavidin with a dopamine transporter-specific ligand, which allowed analysis of the protein in cell edges, filopodia, and lamellipodia for greater understanding of membrane localisation [87].

Another example of NIR IIb QDs is Pb/C/CdS QDs coated with PEG, arginine–glycine–aspartate (RGD) peptides, and catalase. These QDs were seen to accumulate in tumours due to the RGD peptide, allowing for efficient imaging. These QDs were able to increase radiosensitivity due to the lead in the core. This, on top of the killing of cells, caused an increased immune response against the tumour cells leading to immunogenic cell death of the cancer cells. The radiosensitisation was also increased through the use of catalase, which caused the degradation of H_2_O_2_, hence reducing the hypoxic environment of the tumour [88].

QDs have also been used to image macrophages in vivo, ex vivo, and in situ in adipose tissue where the location of macrophages is not well known. These QDs were decorated with a dextran mimetic to target these cells whilst maintaining their biocompatibility. These dextran-mimetic QDs themselves emitted in NIR, as well as being conjugated to radioactive iodine (iodine-124 and iodine-125), allowing for both optical and nuclear imaging. Obese C57Bl/6 mice were then injected with the QDs due to having macrophage-rich adipose tissue and were imaged via PET and CT with the end result of high-resolution imaging of the tissue and macrophages. Due to the vast array of roles macrophages play in the body, such as wound repair, inflammation and immunity, and cancer, this study hoped to develop an agent to analyse macrophages on a cellular level, up to tissues to better understand function [89].

Paramagnetic QD-micelles have been developed for multimodal imaging of angiogenesis. For this, CdSe/ZnS QDs were coated with PEG-DPSE, gadolinium-DTPA-bis(stearylamide) (Gd-DTPA-BSA), and were conjugated with RGD peptide. RGD peptide was used to target integrin receptors which are typically overexpressed in tumour cells. This QD was able to visualise tumour angiogenesis by both fluorescence and MRI in tumour-bearing mice. The QD was able to act quickly, being bound to endothelial cells within 10 min from injection, in a specific manner, and did not undergo extravasation. Angiogenesis was found at the rim of the tumour both by intravital microscopy and by MRI, with fluorescent imaging used to visualise the tumour in the whole animal [90].

QDs have a large potential for use in biological imaging or the cardiovascular and haematopoietic systems via fluorescence and in combination with MRI and X-rays, holding promise for the imaging, diagnosis, and treatment of various diseases such as angiogenesis, tumour vasculature, and atherosclerosis. However, more work is needed to hone these abilities along with ensuring biological compatibility.

## 4. Conclusions

QDs have a vast potential for biological applications, from biosensors to uses within theragnostic agents with many roles, some untapped, in the vascular system. It must be noted that how the QD will be used, i.e., for short term or long-term use, might be critical as long-term use has a greater opportunity for side effects to be induced due to the presence of the QDs within the biological system. Importantly, in this review, we explored the beneficial, potential roles of QDs in the haematopoietic system, from the prevention of plaque growth in atherosclerotic mice models [32], the use of red blood cells in QD chemotherapy [52], and effective imaging of damaged endothelium [8] and HSCs [76]. However, we have also delved into the potential harm and lack of biocompatibility of QDs on the haematopoietic system, from deformation and haemolysis in red blood cells [49], differing findings on the inflammatory responses of leukocytes [59,60], activation of platelets [14,41], and changes to the coagulation cascade [45,46,47]. Given the need of the vascular system for the delivery of QDs to areas of the biological system, further research is greatly needed to establish the differing effects of the different makeups that QDs can have, from shell to ligands, to provide a safe therapeutic or diagnostic agent for use within a medical environment.

## Figures and Tables

**Figure 1 cells-13-00354-f001:**
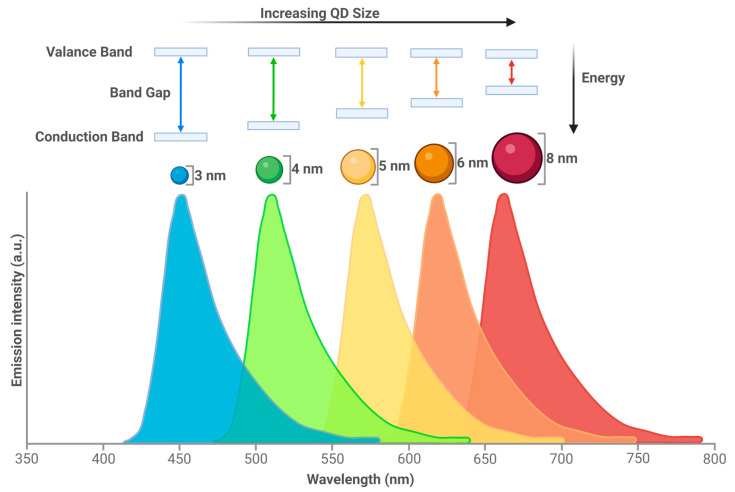
QD size results in emission spectra changes. QDs can fluoresce due to the activity of electrons. As a photon hits a semiconductor, an electron is excited from the valance band (outermost electron orbital of an atom that electrons occupy) to the conduction band (electron orbital that energised electrons can move to). The distance between these two bands is known as the band gap. Narrow band gaps of QDs shift optical properties to lower energies, which result in longer wavelengths [3]. Created with BioRender.com.

**Figure 2 cells-13-00354-f002:**
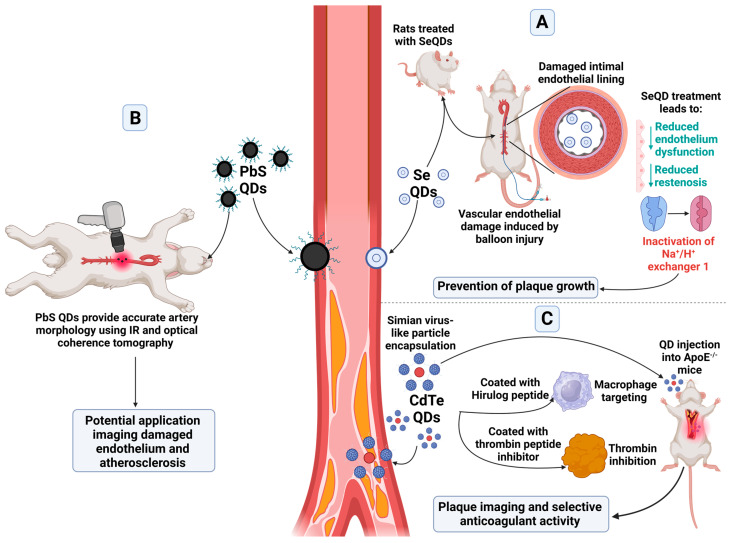
Applications of QDs within the haematopoietic system and atherosclerosis. Select applications of QD types for atherosclerosis can include (**A**) prevention of plaque growth [32], (**B**) imaging of the affected sites including atherosclerotic plaque and damaged endothelium [30], and (**C**) anticoagulant activity and plaque imaging [33]. These applications utilise various QD types, including lead sulphide, cadmium tellurium, and selenium QDs. Created with BioRender.com.

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
