# Peer review of "Quantum Dot Imaging Agents: Haematopoietic Cell Interactions and Biocompatibility"

_cells, 2024, doi:10.3390/cells13040354_

Round 1

Reviewer 1 Report

Comments and Suggestions for Authors

The submitted manuscript describes the applications of quantum dots (QDs) to the haematopoietic system. Although the contents of the submitted manuscript are interesting and attractive to readers, the manuscript structure, especially the order of each chapter, should be reconsidered. The weight of information on the functional material such as QDs and the biological system such as the cardiovascular system is not the same. The reviewer thinks that the authors should note the treatment of chapter 9 (Imaging, Intracellular, and Real Time Imaging), compared to other chapters. Although it is difficult to correctly note this point, the applicable areas of QDs and the biological application to QDs are different. Since there are several strange sentences in the manuscript, the authors should review the manuscript carefully. One example is the final sentence in Page 4. What is 1600 nM?

The authors should pay much attention to Figure. The explanation of Figure 1 is not sufficient. What is the vertical axis? There is no specific explanation on the “Band Gap” including “Valance Band” and “Conduction Band” and the relationship between the particle size such as 3 nm and the fluorescence color. Figure 2 is too busy. A clearer illustration is desirable.

Comments on the Quality of English Language

It is generally well-written.

Author Response

Reviewer 1

The submitted manuscript describes the applications of quantum dots (QDs) to the haematopoietic system. Although the contents of the submitted manuscript are interesting and attractive to readers, 

We thank the reviewer for this comment and agree that this is an area of interest

The manuscript structure, especially the order of each chapter, should be reconsidered. The weight of information on the functional material such as QDs and the biological system such as the cardiovascular system is not the same. The reviewer thinks that the authors should note the treatment of chapter 9 (Imaging, Intracellular, and Real Time Imaging), compared to other chapters. Although it is difficult to correctly note this point, the applicable areas of QDs and the biological application to QDs are different.

We have reconsidered where a number of different parts have been located within the manuscript.  This has led to an alteration in the weighting of certain chapters, with the last chapter especially being reduced in size (and renamed as QD Imaging of Haematopoietic cells) as parts have been moved to other areas of the review.  This has helped to ensure that the structure of the review has been improved.

We have also changed the title of the review to more emphasise that we are looking at the interactions with blood cells and QD biocompatibility, which is much more the focus of our review article.

Since there are several strange sentences in the manuscript, the authors should review the manuscript carefully. One example is the final sentence in Page 4. What is 1600 nM?

The manuscript has been thoroughly reviewed and all changes have been tracked for the benefit of the reviewers. There were a number of unit errors within the original manuscript which have been corrected.

The authors should pay much attention to Figure. The explanation of Figure 1 is not sufficient. What is the vertical axis? There is no specific explanation on the “Band Gap” including “Valance Band” and “Conduction Band” and the relationship between the particle size such as 3 nm and the fluorescence color. Figure 2 is too busy. A clearer illustration is desirable.

We thank the reviewer for this comment. Both Figure 1 and Figure 2 have been altered to improve their clarity. Figure 1 has an improved figure legend, with additional information to understand in the figure legend, and a clear label for the y axis.  Figure 2 has been made less busy as requested.

Reviewer 2 Report

Comments and Suggestions for Authors

The topic of the review is novel, however, the structure is not good, and figures are not enough, in addition, so many flaws need to be revised. Please refer the following comments.

Major:

1.  The structure is not reasonable since Cardiovascular System and Hematopoietic System are parallel, so it’s proper to rearrange part 3-8 as part 3.1-3.6, and add a new total title of “Hematopoietic System” as part 3.

2.  Figures or schemes should be inserted in part 2 “2. Cardiovascular System” and part 3 “3. Hematopoietic System” to show more details of corresponding works since they are the main part of the review.

3.  The authors mentioned “Studies into the effects of InP/ZnS QDs on platelets have also been conducted. These QDs, due to the lack of heavy metal, are seen to be a biologically safer alternative to Cd QDs”, however, “In” should be a kind of heavy metal since its atom number is even larger than Cd. 

Minor:

1.  First letter of “Indium Phosphide QDs”.

2.  Delete dysfunction from “CVD dysfunction”.

3.  Comma after “The larger the size”.

4.  “reactive oxygen species [8-12]” should be “reactive oxygen species (ROS) [8-12]” since it appears for the first time.

5.  Format of “-NH2”.

6.  Whole name of “IR” when it appears first time, the same as “VWF”, “EGFR”, “AMPA”, “FRET”, and so on, please check carefully.

7.  “nM" should be “nm” when it means “nanometer”this issue appears so many times! A blank should exist between “800nm” and “60fps”.

8.  “in vivo” should be italic.

9.  150-400*106/ml, why not change as 1.5-4*108/ml?

10. “amine functionalized” should be “amine-functionalised”.

11. Usually the lifespan of red cells should be considered as 4 month (120 days) as mentioned in numerous text book and literatures (like this: https://link.springer.com/article/10.1186/s12951-023-02060-5 ), not 90 days.

12. “two times lower” is not a scientific statement, should replace with “half”.

13. “The rout ofshould be “The route of”, correct?

14. part 9 “9. Imaging, Intracellular, and Real Time Imaging” should be “9. Intracellular Imaging and Real Time Imaging”?

15. Is “obese mice” a kind of mice? I doubt it.

16. Keep English format the same, all use British English or American English, I think you are using British English mainly.

17. “hone” should be “home”, correct?

18. It’s not accurate to say “QDs have a vast potential for biological applications, from biosensors to chemotherapeutic agents” since QD is not a kind of chemotherapeutic agents, but it can load chemotherapeutic agents, the same as in “QD chemotherapy”.

Comments on the Quality of English Language

Many spell problems, please see comments part.

Author Response

Reviewer 2

We thank the reviewer for their helpful comments and have restructured the review in response to their useful comments.

Major:

  1. The structure is not reasonable since Cardiovascular System and Hematopoietic System are parallel, so it’s proper to rearrange part 3-8 as part 3.1-3.6, and add a new total title of “Hematopoietic System” as part 3.

This has been completed.  We have also altered the title of the review to reflect the overall emphasis of the review.

  1. Figures or schemes should be inserted in part 2 “ Cardiovascular System” and part 3 “3. Hematopoietic System” to show more details of corresponding works since they are the main part of the review.

The figures have been updated to ensure their clarity.

  1. The authors mentioned “Studies into the effects of InP/ZnS QDs on platelets have also been conducted. These QDs, due to the lack of heavy metal, are seen to be a biologically safer alternative to Cd QDs”, however, “In” should be a kind of heavy metalsince its atom number is even larger than Cd. 

Heavy metal is a term that has multiple meanings dependent on the field of Chemistry, physics or metallurgy.  To avoid confusion this has been rewritten to emphasise the increased biocompatibility of the InP /ZnS QDs in comparison to Cadmium QDs.

Minor:

  1. First letter of “Indium Phosphide QDs”.

This has been completed

  1. Delete dysfunction from “CVD dysfunction”.

This has been completed

  1. Comma after “The larger the size”.

This has been completed

  1.  “reactive oxygen species [8-12]” should be “reactive oxygen species (ROS) [8-12]” since it appears for the first time.

This has been completed

  1. Format of “-NH2”.

This has been completed

  1. Whole name of “IR” when it appears first time, the same as “VWF”, “EGFR”, “AMPA”, “FRET”, and so on, please check carefully.

This has been completed

  1. “nM" should be “nm” when it means “nanometer”,this issue appears so many times! A blank should exist between “800nm” and “60fps”.

Yes we apologise for the mistake within the unit used.  This has been corrected in all places, and a space also put in between number and unit.

  1.  “in vivo” should be italic.

This has been completed

  1. 150-400*106/ml, why not change as 1.5-4*108/ml?

Thank you for this comment.  On reflection we have altered this value to the clinically relevant value of 150-400x109/L.  We feel this would be most appropriate here.

  1. “amine functionalized” should be “amine-functionalised”.

This has been completed

  1. Usually the lifespan of red cells should be considered as 4 month (120 days) as mentioned in numerous text book and literatures (like this: https://link.springer.com/article/10.1186/s12951-023-02060-5 ), not 90 days.

Thank you for his comment this has been corrected

  1. “two times lower” is not a scientific statement, should replace with “half”.

This has been completed

  1. “The rout of” should be “The route of”, correct?

This has been completed

  1. part 9 “9. Imaging, Intracellular, and Real Time Imaging” should be “9. Intracellular Imaging and Real Time Imaging”?

This section has now been changed to imaging of haematopoeitic cells, as the information has been rearranged.

  1. Is “obese mice” a kind of mice? I doubt it.

We have now changed this to Obese C57BL/6 mice for clarity.

  1. Keep English format the same, all use British English or American English, I think you are using British English mainly.

Yes we are using British English.  This has been completed

  1. “hone” should be “home”, correct?

We have changed Hone to improve for clarity.

  1. It’s not accurate to say “QDs have a vast potential for biological applications, from biosensors to chemotherapeutic agents” since QD is not a kind of chemotherapeutic agents, but it can load chemotherapeutic agents, the same as in “QD chemotherapy”.

We have altered the text to say QDs can be included within theragnostic agents, rather than directly indicating that they are chemotherapeutic agents.

Round 2

Reviewer 2 Report

Comments and Suggestions for Authors

The authors solved my concerns, and I have no more comments.